# Drug company payments to General Practices in England: Cross-sectional and social network analysis

Eszter Saghy[1]☯, Shai Mulinari[2]☯, Piotr Ozieranski[3]☯ *

1 Faculty of Pharmacy, Division of Pharmacoeconomics, University of Pecs, Pecs, Hungary, 2 Department of Sociology, Lund University, Lund, Sweden, 3 Department of Social and Policy Sciences, University of Bath, Bath, United Kingdom

☯ These authors contributed equally to this work.
* po239@bath.ac.uk

**Data Availability Statement:** All excel files including the data we used in this research are available from Figshare data repository (doi: https://doi.org/10.6084/m9.figshare.14787186.v1).

**Funding:** This study (SM as PI and PO as Co-I) was supported by the grant 'What can be learnt

## Abstract

Although there has been extensive research on pharmaceutical industry payments to healthcare professionals, healthcare organisations with key roles in health systems have received little attention. We seek to contribute to addressing this gap in research by examining drug company payments to General Practices in England in 2015. We combine a publicly available payments database managed by the pharmaceutical industry with datasets covering key practice characteristics. We find that practices were an important target of company payments, receiving £2,726,018, equivalent to 6.5% of the value of payments to all healthcare organisations in England. Payments to practices were highly concentrated and specific companies were also highly dominant. The top 10 donors and the top 10 recipients amassed 87.9% and 13.6% of the value of payments, respectively. Practices with more patients, a greater proportion of elderly patients, and those in more affluent areas received significantly more payments on average. However, the patterns of payments were similar across England's regions. We also found that company networks–established by making payments to the same practices–were largely dominated by a single company, which was also by far the biggest donor. Greater policy attention is required to the risk of financial dependency and conflicts of interests that might arise from payments to practices and to organisational conflicts of interests more broadly. Our research also demonstrates that the comprehensiveness and quality of payment data disclosed via industry self-regulatory arrangements needs improvement. More interconnectivity between payment data and other datasets is needed to capture company marketing strategies systematically.

## Introduction

Drug company payments to the healthcare sector can create conflicts of interest biasing clinical practice [1], research [2,3], and policymaking [4]. A key global trend towards addressing this risk involves payment disclosure via either public regulation (e.g., the US Open Payments or

from the new pharmaceutical industry payment disclosures?' awarded by the Swedish Research Council for Health, Working Life and Welfare (FORTE), no. 2016-00875 and the grant 'Following the money: cross-national study of pharmaceutical industry payments to medical associations and patient organisations', awarded by The Swedish Research Council (VR), no. 2020-01822. ES's work was supported by the former grant. The funders had no role in study design, data collection and analysis, decision to publish, or preparation of the manuscript.

**Competing interests:** PO's PhD student was supported by a grant from Sigma Pharmaceuticals, a UK pharmacy wholesaler and distributor (not a pharmaceutical company). The PhD work funded by Sigma Pharmaceuticals is unrelated to the subject of this paper. SM's partner is employed by PRA Health Sciences, a global Contract Research Organization whose costumers include many pharmaceutical companies. ES has no conflicts of interest to declare. This does not alter our adherence to PLOS ONE policies on sharing data and materials.

French Transparence Sante databases [5]) or industry self-regulation (e.g., most European countries, including the UK [6], Japan [7], and Australia [5]).

Research on payment disclosures has centered on *individual healthcare professionals* [8–11], with increasing evidence from the US of even small payments influencing drug prescription [12–14] and healthcare cost [14,15]. However, *healthcare organisations* (HCOs), including service providers, regulators or medical societies, have received less attention, even though they shape healthcare delivery via resource allocation, regulatory decisions, recommendations and guidelines [16,17]. The limited interest in payments to HCOs in the US [18] seems to reflect the fact that the Sunshine Act only covers payments to hospitals. However, the definition of organisational-level recipients adopted in European countries with self-regulation is considerably broader, therefore allowing for capturing the unique compositions of HCOs in national healthcare systems [21]. Additionally, pharmaceutical companies and trade groups typically do not interpret payments to HCOs as falling under European data privacy laws, which prevents these recipients from refusing to have their payments disclosed [17]. This contrasts with payments to healthcare professionals, interpreted by the industry as "personal data", and therefore characterised by pervasive non-disclosure, precluding comprehensive analysis [11]. For example, in 2015 in the UK, only around 50% of the disclosed payments had any information about the individual recipients [10], and this had increased only to about 60% by 2019 [11].

Despite the relatively greater data availability, the UK is the only European country with patterns of payments to HCOs described at the national level [19] and in relation to organisations commissioning (or procuring) healthcare services for patients [20,21] as well as secondary-care providers [22]. Building on this research, we examine payments to General Practice (GP) surgeries (henceforth, practices), excluding specialist practices providing services related to specific fields of medicine [23]. As healthcare is organised differently across the UK [24], we examine England as its largest part. We focus on practices given their vital role in healthcare delivery in England, with over 60 million patients being registered at practices [25] and over 300 million appointments annually, compared to 23 million accident and emergency service hospital visits [26]. Further, over half of the total National Health Service (NHS) pharmaceutical spending involved prescriptions issued by practices [26].

We anticipate that practices will be a key target of company payments. Consistent with patterns of payments to HCOs in the UK [19] and the US [18], we also expect a few companies and recipients to concentrate most payments. Furthermore, following US research emphasising the importance of relatively small payments in influencing physicians [18,27,28], we predict that most companies will make many relatively small payments rather than a few large payments.

We also consider key practice characteristics–location, size, and some features of the patient population–as potentially affecting company choices about who receives payments. We hypothesise that the *proportion* of practices receiving payments is roughly equal across the regions of England. However, we expect to see differences in the *amount* of payments between practices in different parts of England reflecting previously demonstrated regional variation in prescribing patterns [29]. We anticipate more payments to practices with a higher number of registered patients compared to those with a smaller clientele, given the predicted greater "return on investment" for companies. Specifically, we expect that practices with higher shares of patients over the age of 65 will receive more payments compared to those with fewer elderly patients, due to, for example, the greater tendency for polypharmacy in elderly populations [30]. Moreover, we expect practices in more deprived locations to obtain more payments compared to those in more affluent areas [31] as, for example, studies in Northern Ireland [32] and Scotland [33] found greater numbers of prescriptions per patient in the most deprived areas

compared with the least deprived ones, making them potentially more attractive as payment recipients.

Given the well-documented "relational" nature of the pharmaceutical industry, including its attempts to develop ties to, and visibility among, actors seen as vital for driving product uptake and profitability [34–37], we use social network analysis (SNA) to explore social structures involved in making payments to practices. Drawing on emerging applications of SNA to study pharmaceutical industry payments and marketing [38], we anticipate that connections established by making payments to practices are not accidental. For example, data analytics companies have offered SNA insights to map Key Opinion Leaders (KOLs) in the medical field in the US [38–41], and it is likely that similar services are also used in European countries [40,41]. As data disclosed within industry self-regulation has no information on products related to payments [6,19], SNA cannot trace product competition among companies. Instead, we examine i) which companies are interested in making payments to the same practices, ii) which companies are dominating the payment networks (if any), and iii) the differential density of connections within such networks. In so doing, we consider two types of networks. We interpret networks based on the *value* of payments as indicating the "importance" of a practice for a drug company, while networks involving the *number* of payments as pointing to the intensity of interactions with the practice. More frequent payments may, for example, enhance a company's visibility, which could be an important goal of marketing efforts [42].

We had two specific objectives. We sought to analyse, first, the distribution of and factors associated with payments across drug companies and practices in England; and, second, the structure of connections between drug companies established by making payments to the same practices.

## Methods

### Study design

Our study combines cross-sectional and SNA analysis of drug company payments to practices in England. We combined Disclosure UK [43]–an annually published dataset including, among others, non-research payments to named HCOs, disclosed by companies following the Code of Practice of the Association of the British Pharmaceutical Industry (ABPI) [19]–with separately sourced information on practice characteristics. We analysed the distribution of payments across practices and companies and assessed the associations with selected practice characteristics. We then mapped the structure of connections between companies and their shared practices using SNA. Given the exploratory nature of our research, which involved combining datasets which had not been previously used to analyse drug company payments, our study did not follow an a priori protocol.

### Data sources and extraction

We extracted data on company payments to practices from the 2015 edition of Disclosure UK as this is the only one for which previous research categorised payment recipients, which enabled isolating payments to practices [19]. The relationship between practices and the previously examined larger category of public sector primary care providers [19] is explained in S1 Appendix. To prevent any payments to practices from being unnecessarily excluded due to companies potentially misidentifying their ultimate recipients [19], we combined payments practices identified in either "Institution name" or "Institution location" columns of the dataset (2,945 payments in total, out of which 2,747 were used for the analysis). The section of Disclosure UK we analysed (Online supplement 1) can be matched with Disclosure UK version 20160630. To allow accurate comparison of payment values between companies we adjusted

them for VAT using information from company "methodological notes", as described elsewhere [19].

We used the GP Friends and Family Test (FFT) dataset [44] to assign unique codes to practices identified in Disclosure UK. The practice names were matched with the practice codes based on comparing practice names and addresses from the two datasets. For each practice with a unique code we obtained the number of registered patients using the Patients Registered at a GP Practice 2015 NHS dataset [25]. We also used this dataset to calculate the share of patients over 65. In addition, we obtained multiple deprivation index (MDI) decile scores for the postcode of each practice from the website of the Ministry of Housing, Communities and Local Government [45]. The MDI is an aggregated score of 37 indicators providing information on income; employment; health and disability; education, skills and training; crime; barriers to housing and services and living environment [46]. We divided practices into 4 quartiles based on their MDI score.

The categorisation and cleaning of drug company payment data is described elsewhere [42]. The extraction of data from the datasets with practice characteristics is described in the protocol available in S2 Appendix.

## Analysis

**Statistical analysis.** We used R [47] version 1.4.1717 to analyse the distribution of payments descriptively and to assess differences across selected practice characteristics. As the distribution was heavily skewed, we examined medians and interquartile ranges. Significance of the difference in the value of payments between different groups was assessed using the Wilcoxon nonparametric statistical test. The reference groups are London in regional comparison; Lowest number of patients ($1^{st}$ quartile) in practice size comparison; Lowest share of elderly patients ($1^{st}$ quartile) in elderly patient population comparison, and Most deprived ($1^{st}$ quartile) in MDI comparison. The threshold for significance (alpha) was set to 0.05.

**Social network analysis.** We first created company by practice matrices in MS Excel, then converting them into company by company matrices to allow for examining connections between companies established by "shared" practices, i.e., practices to which any pair of companies made payments. We report findings calculated based on "valued" matrices, with the number of shared practices shown at the intersect of companies. We created separate matrices for different thresholds of the number and value of payments involved in establishing connections between companies; quartiles of the overall number and value of payments per company; and practice characteristics (i.e., quartiles of the total number of patients; quartiles of the share of patients over 65; and quartiles of the MDI of the postcodes in which the practices were based) (Online supplement 2).

We analysed the matrices in UCINET version 6.689 [48], visualising them in Gephi version 0.9.2. [49]. We calculated each company's *centrality*, which is the number of ties a company has, i.e. the number of connections to other companies established by making payments to the same practices [50]. We also calculated network *centralisation*, showing, on a scale from 0 to 1, the extent to which a network is dominated by one company [50]. Centralisation score is measured as the ratio of the actual sum of centrality score differences and all possible sum of centrality score differences [51]. Finally, we calculated network *density*–the strength of existing ties between actors as a share of all possible ties. In our valued networks, density is calculated by dividing the sum of shared practices between all companies in a network by the total of all possible connections [52]. We report findings relating to networks established based on the *value* of payments made by drug companies but throughout the results we also signpost to web appendices with additional findings relating to networks considering the *number* of payments.

Overall, from the SNA perspective, we expect to be able to detect companies dominating the field of payments to practices and patterns of lower and higher number of shared practices based on different payment sizes and practice characteristics mentioned above.

# Results

## Descriptive analysis of the distribution of payments to practices

In total, 37 drug companies made 2,945 payments, worth £ 2,726,017.77 to 1,790 practices. In 2015, these companies represented 37.0% of those reporting payments to HCOs in England. Consistent with our expectations regarding the importance of practices as a target of industry payments, payments to practices constituted 6.5% of the value of all payments made to HCOs in England (S3 Appendix) and practices ranked 5th of all HCOs receiving the highest amount of payments after universities, NHS foundation trusts, NHS trusts, and multi-professional organisations.

We excluded from further analysis 198 payments (6.72%), worth £166,351.74 (6.1%) made to 147 (8.21%) practices as we could not link them to practice codes. These practices were randomly distributed across England (S4 Appendix), with the median payment values similar to those in the rest of the dataset. Our final sample, therefore, comprised 2,747 payments worth, £2,559,666.03, made by 34 companies to 1,643 practices. These payments were for donations and grants (76.36%), contributions to costs of events (22.51%), and fees for service and consultancy (1.14%) (S5 Appendix).

As expected, payments were highly concentrated (Table 1). Although three-quarters of practices received no more than two, the top practice received as many as 132. Most companies were "small donors", with three-quarters making no more than 81 payments, but the maximum number was almost a thousand (i.e., Bayer). The value of payments was similarly concentrated. Although three-quarters of practices received no more than £1,5k, the top one accumulated almost ten times more. Likewise, while three-quarters of companies made payments worth no more than £100k, those made by the top donor were worth more than 7.5 times as much. Three-quarters of companies made payments to no more than 56 practices, but the top donor, Bayer, remarkably, made its 998 payments to 778 practices (2.47% of the value of Bayer's payments were contributions to costs of events, 96.19% were donations and grants, and 1.34% were fees for service and concultancy). A majority of practices only received payments from one company, while the the top recipient received payments in total from 18 companies.

Table 2 further evidences the concentration of payments, with those made by the top ten donors constituting, respectively, 93.64% and 83.69% of the total number and value of

**Table 1. Summary of drug company payments to general practices.**

| Level of analysis | Minimum | Median [IQR] | Maximum |
|---|---|---|---|
| Single payment value (£) | 8.00 | 320.00 [170.00–869.00] | 49,420.80 |
| Value of payments per general practice (£) | 9.59 | 576.00 [217.25–1,520.75] | 148,395.20 |
| Number of payments per general practice | 1.00 | 1.00 [1.00–2.00] | 132.00 |
| Value of payments per company (£) | 80.00 | 9,036.00 [1,003.00–97,377.00] | 765,987.77 |
| Number of payments per company | 1.00 | 14.50 [3.25–80.75] | 998.00 |
| Number of practices per company | 1.00 | 8.50 [3.00–56.00] | 778.00 |
| Number of companies per practice | 1.00 | 1.00 [1.00–1.00] | 18.00 |

Notes: This table is based on drug company payments reported in Disclosure UK (2015, version 20160630).

**Table 2. Payments made by the top 10 drug company donors to general practices.**

| Company | Total value of payments (£) | Total number of payments | Number of practices paid | Median value of single payments (£) [IQR] |
|---|---|---|---|---|
| Bayer | 765,987.77 | 998 | 773 | 434.50 [217.20–869.00] |
| Pfizer | 360,556.90 | 140 | 105 | 1,412.10 [236.00–3,907.00] |
| Eli Lilly | 271,139.00 | 260 | 185 | 200.00 [168.00–3,353.00] |
| Sanofi Aventis | 269,965.82 | 149 | 126 | 1,000.00 [240.00–2,400.00] |
| AstraZeneca | 153,865.25 | 85 | 16 | 250.00 [150.00–550.00] |
| Boehringer Ingelheim | 145,070.58 | 213 | 146 | 392.00 [177.60–640.00] |
| Merck Sharp & Dohme | 124,062.80 | 63 | 50 | 800.00 [195.80–4,400.00] |
| Takeda | 112,428.80 | 94 | 58 | 240.00 [180.00–921.40] |
| Napp | 97,743.39 | 235 | 212 | 38.49.00 [28.90–111.73] |
| Servier | 96,162.40 | 62 | 61 | 1,567.00 [576.00–1,567.00] |

Notes: This table is based on Disclosure UK (2015, version 20160630).

payments made to practices. Of the 10 companies, Bayer was dominant in the number and value of payments, as well as the number of practices to which payments were made. A similar table including the top 10 recipients is presented in S6 Appendix.

The evidence of companies' preference for small payments was mixed. Most payments were indeed relatively small, with three-quarters being no more than £869.0 (Table 1). However, important differences existed among the biggest donors (Table 2). Despite the varying overall size of payments per company, the comparison of median payment values shows that companies such as Pfizer, Merck Sharp & Dohme, and Servier made fewer but more substantive payments, while Bayer, AstraZeneca and Napp made a larger number of smaller payments. However, the comparison of the median values at the payment and practice levels suggests there were two groups of companies prioritising small payments, with one concentrating on a smaller number of practices, while the other dispersing its payments across a larger number of practices. For example, while AstraZeneca made 85 payments to 16 practices, Servier made 62 payments to 61 practices. This reflects the overall payment distribution (Table 1), with three-quarters of practices receiving no more than two payments.

The relationships between payment patterns and practice characteristics were broadly consistent with our expectations (Table 3). In most regions of England, the shares of practices receiving payments ranged between a fifth and a quarter of all practices. The only two regions with markedly lower shares were London (8.88%) and North East England (14.66%). Nevertheless, the number of practices receiving payments varied considerably between the regions, with only 107 located in North East England and 267 in North West England. The median payment values also displayed regional differences, with practices in North East England having the median value almost twice as high as those in London and South East England (London vs. North East England; p <0.001). Moreover, the median value of payments per practice increased together with the practice size and the proportion of patients over 65 (all other quartiles are significantly different from the first quartile). However, unexpectedly, practices in the most deprived areas (1st quartile based on MDI) received significantly smaller payments than other practices.

## Social network analysis of connections between drug companies

Fig 1 shows valued networks of connections between companies making payments to the same practices. A connection indicates at least one payment made to the same practice and line thickness and darker colour correspond with a greater number of shared practices. Therefore,

**Table 3. Breakdown of drug company payments according to general practice characteristics.**

| Classification | Group | Median value of payments (£) [IQR] | P-value | Number of general practices receiving payments (% out of total practices in the region) |
|---|---|---|---|---|
| | London | 434.50 [217.25–2,600.00] | Ref | 140 (8.88%) |
| Regional breakdown | East Midlands | 434.50 [217.25–869.00] | 0.756 | 147 (19.57%) |
| | East of England | 600.00 [208.63–1,086.25] | 0.994 | 136 (19.26%) |
| | North East England | 869.00 [434.50–2,909.12] | <0.001 | 107 (14.66%) |
| | North West England | 665.88 [217.25–3,168.95] | <0.001 | 261 (19.30%) |
| | South East England | 434.50 [182.04–910.70] | 0.086 | 249 (26.57%) |
| | South West England | 651.75 [320.00–1,104.00] | 0.221 | 168 (23.90%) |
| | West Midlands | 461.42 [164.00–1,344.00] | 0.826 | 220 (23.63%) |
| | Yorkshire and the Humber | 460.00 [217.23–2,422.12] | 0.243 | 215 (no data) |
| Breakdown based on number of registered patients | Lowest number of patients (1st quartile) | 217.25 [82.00–245.59] | Ref | |
| | Lower number of patients (2nd quartile) | 434.50 [200.00–651.75] | <0.001 | |
| | Higher number of patients (3rd quartile) | 869.00 [486.50–2,175.00] | <0.001 | |
| | Highest number of patients (4th quartile) | 2087.20 [1,012.00–4,400.00] | <0.001 | |
| Breakdown based on share of patients over 65 years | Lowest share of elderly patients (1st quartile) | 320.00 [167.50–863.12] | Ref | |
| | Lower share of elderly patients (2nd quartile) | 585.00 [217.25–1,409.56] | <0.001 | |
| | Higher share of elderly patients (3rd quartile) | 651.75 [242.50–1,864.25] | <0.001 | |
| | Highest share of elderly patients (4th quartile) | 869.00 [434.50–2,283.12] | <0.001 | |
| Breakdown based on index of multiple deprivation | Most deprived (1st quartile) | 434.50 [200.00–1,157.68] | Ref | |
| | More deprived (2nd quartile) | 587.70 [217.25–1,470.70] | 0.041 | |
| | Less deprived (3rd quartile) | 651.75 [217.25–1,699.80] | 0.001 | |
| | Least deprived (4th quartile) | 651.75 [325.75–2,056.00] | <0.001 | |

Notes: The share of the number of practices out of the total were only included for the regional breakdown because data could only be extracted for this variable. We did not find data on the number of practices in Yorkshire and the Humber. These practices are possible counted together with practices in North East England. Significance of the difference in the value of payments between different groups was assessed using Wilcoxon nonparametric statistical test. Reference groups are London, Lowest number of patients (1st quartile), Lowest share of elderly patients (1st quartile), and Most deprived (1st quartile). This table is based on Disclosure UK (2015, version 20160630), the GP Friends and Family Test (FFT) dataset, and the Patients Registered at a GP Practice 2015 NHS dataset.

companies connected with thicker and darker lines can be interpreted as having a shared interest in a greater number of practices. We demonstrate configurations of companies at different level of the value of payments. In Fig 1A, all companies are shown, while in Fig 1B–1D only companies making individual payments worth at least £100, £1000, and £2,500, respectively, are shown.

As the value of payments increased, the number of companies decreased from 29 (Fig 1A) to 11 (Fig 1D), suggesting that only a few companies engaged with practices using high-value payments. The configurations of companies also changed, indicating similarities and differences in how they engaged with practices with payments above a certain value.

Fig 1A–1D can also be analysed in terms of their density, where higher density of a graph means stronger connections between a greater proportion of companies. The density of the graphs decreases from A to D, as the value of payments increases (see density scores in S7 Appendix). This suggests that there is an overall lower interest in the same practices as

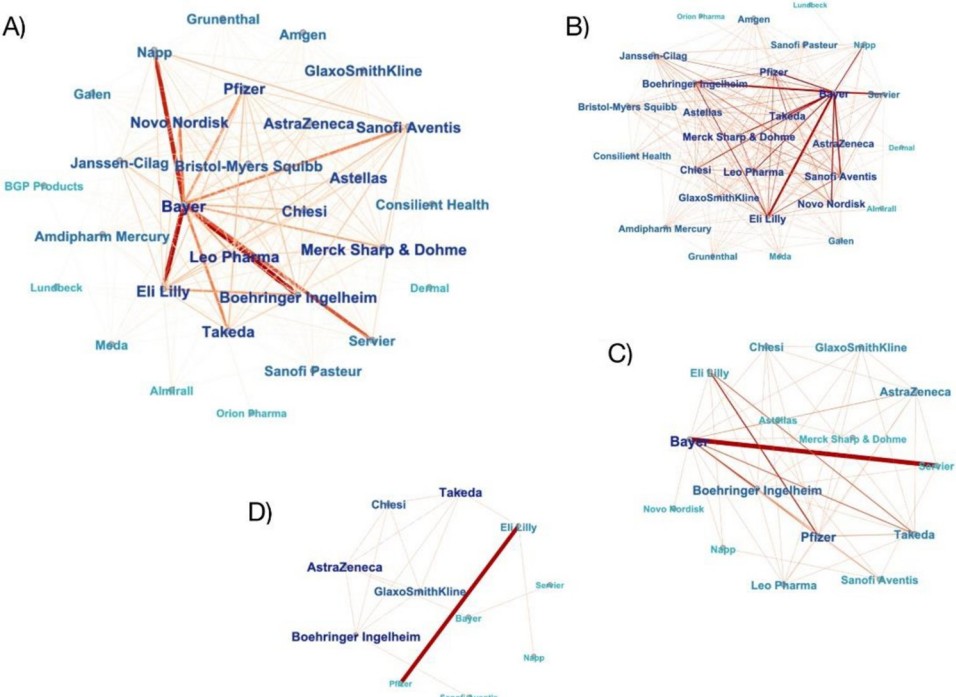

**Fig 1. Networks based on the value of payments.** Notes: 1A) network of all payments; 1B) network of payment over £100 per practice; 1C) network of payment over £1000 per practice; 1D) network of payment over £2500 per practice. Fig 1A–1D shows the visualisation of networks based on the value of payments, created in Gephi. Node label size and darkness corresponds to the centrality of a company, the strength and darkness of the lines corresponds to the number of shared practices between companies. The networks visibly change as the payment number to a single practice increases.

expressed by higher value payments, which means that one practice does not usually receive high value payments from multiple companies.

Moving on to specific companies, considering all payments, Bayer had the greatest shared interest in practices with Napp, Eli Lilly and Boehringer Ingelheim (Fig 1A). But when only accounting for payments worth over £1,000, Bayer made the highest number of payments to the same practices with Servier. In addition, among companies making the highest-value payments, over £2,500, those with the greatest shared interest in practices were Eli Lilly and Pfizer (Fig 1D).

This trend corresponds with companies' centrality scores, which rose with the increasing number of connections with other companies. In Fig 1A–1C, Bayer is the company with the highest centrality score, while in Fig 1D Eli Lilly is the most central one of the network (see centrality scores in S7 Appendix).

The centralisation level of a graph indicates the extent to which one company dominates a network by being connected to a high number of companies, while other companies have less connections. From the four networks, Fig 1A is the most centralised, with Bayer dominating the network. This means that on many occasions when a company makes payments to a practice, Bayer also makes a payment there. Similarly to density, centralisation decreases as the value of payments increases (see centralisation scores in S7 Appendix). Further graphs of networks based on the number of payments can be found in S8 Appendix.

In S9 Appendix, we present additional results for valued drug company networks associated with making payments to practices with different characteristics. Interesting differences can be

observed in terms of centralisation (the extent to which one company dominates a network) and centrality (the number of connections a company has), which we are reporting here, while differences between networks in density are not substantial. Regarding the region in which the practices were located, the highest centralisation was observed in South West of England, while the lowest–in South East England. Across the nine regions of England, Bayer had the highest centrality scores, with only Eli Lilly matching it in London and South East England. In company networks established based on making payments to practices of different sizes, a trend existed of increasing network centralisation as the number of patients increased, with the highest centralisation score in the third quartile of practice size. Bayer was, again, the most central company in all four (quartile 1 –quartile 4) networks showing payments made to practices with different patient numbers. In the networks involving payments made to practices based on the proportion of patients over the age of 65, the centralisation score does not change substantially with the increase in the proportion of elderly patients. Bayer also remained the company with the highest centrality score in all quartiles. A similar trend exists in networks with the MDI index (1$^{st}$ quartile being the most deprived) of the location of the practice. Bayer was, yet again, equally dominant in the most and least deprived areas.

## Discussion

As far as we know, this is the first study examining drug company payments to the primary care sector. We find that general practices were a major target of industry payments in England, placing them in the top five and two of organisational recipients based on the value and number of payments, respectively (S3 Appendix). While the value of payments received by general practicioners is unknown given the big gaps in individual-level payment data, they could exceed considerably the organisational-level payments to practices [10]. Notably, the payments made to practices in England (£2.7m) were almost twenty times lower than those made to individual healthcare professionals in UK in 2015 (£50.9m) [10]. Overall, our findings suggest that more attention is needed to drug company payments to organisations and to organisational conflicts of interest [53–55].

Turning to the modes of financial engagement with practices, the high value of "grants and donations" (almost 65%) suggests that companies often provided them with "medical and educational goods and services", which may bear company names but not product names [56]. Contrastingly, the low value of consultancy payments (less than 2%) suggests limited scope of practices offering, on behalf of their employees, services such as "market research" (defined broadly as "the collection and analysis of information" on medicines) or "chairing and speaking at meetings, assistance with training and participation in advisory boards" [56]. "Contributions to cost of events", accounting for around a third of payments, covers events, such as conferences, organised by practices or third parties on their behalf.

Payments to practices were highly concentrated, just like in the UK overall [19]. From the industry side, more than a third (37%) of all companies making payments to HCOs in England reported having made payments to practices. Only a few companies were big donors, with the payment landscape largely dominated by one company, Bayer, which, incidentally, was also identified as the second largest source of payments to healthcare professionals in the UK in 2015 [10]. Bayer was dominanant across all regions of England, practice sizes, and patient population profiles. The SNA provided further evidence of concentration of payments among companies.

We also saw concentration of payments among practices, with many receiving only small or occasional payments, yet with a narrow subset being heavily exposed to industry funding. Although the conference or education budgets of the top recipients are unknown, the volume

of reported payments suggests that the industry–or, indeed, specific companies–were a major source of such support. This is important as research on drug company funding within the healthcare sector [5], including patient organisations [57], highlights risks associated with dependency on industry funding, especially coming from a few donors.

Not only have we found significant regional differences in payment values received by practices across England, but we have also revealed that practices with the lowest number of patients, the lowest share of elderly patients, and those in the most deprived areas receive significantly lower amount of payments. Why practices in most most deprived areas recive less industry funding, and the consequence of this for general practicioners and their patients, should be investigated further.

We identified some evidence of the high-frequency but low-value payment strategy, which has been highlighted as potentially instrumental in generating networks of obligation with US healthcare professionals [58–62]. Here, we unearthed some divergence within this strategy, with some companies making many small payments to different practices, while others concentrating their small payments on fewer practices.

While these differences may indicate contrasting marketing strategies, our interpretation is constrained by the absence of information on products related to payments. Therefore, unlike with meals and small gifts reported in relation to US physicians [6,19,28,59], we do not know the significance of "small" payments, for example, for establishing extended reciprocity at the organisational level. Investigation of payment strategies would be *even less* possible in other European countries with self-regulation of payment disclosure. This is because the ABPI is the only European pharmaceutical industry trade group mandating its member companies not to aggregate payments to HCOs annually per recipient, which allows comparing payments of different sizes. While we are not aware of any detailed guidance from the ABPI associated with this requirement, a review of cases from the UK drug industry self-regulatory authority, the PMCPA, has not identified any relevant compliants. Therefore, it is unlikely that companies had difficulties in interpreting how payments to HCOs should be itemised.

While the key issue of company marketing cannot be addressed directly in the European self-regulatory context, previous research on European self-regulatory systems has captured companies' marketing *indirectly* by considering the nature and frequency of investigations into unethical marketing for specific products, highlighting heavy marketing of drugs prescribed in general practice—antidepressants in the late 1990s [63], followed by anti-diabetics and urologics (mainly erectile dysfunction drugs) in the next decade [64]. Similarly, we note that between 2012–2018, Bayer was sanctioned by the PMCPA [65,66] on no less than 12 occasions for unethical marketing of Xarelto (rivaroxaban), a direct oral anticoagulant (DOAC) often prescribed by general practicioners as stroke prophylaxis in patients with atrial fibrillation [67], suggesting that Bayer's payments to practices could be associated with this drug. Indeed, DOACs have been identified as heavily marketed products in the UK [68].

Finally, the prominence of drug company payments does not seem to be matched adequately by governance frameworks available to practices. Although NHS England requires NHS trusts and clinical commissioning group employees to record externally sponsored events and urges NHS staff to decline gifts that may affect their professional judgement [69], less clarity exists regarding organisational conflicts of interests, which might be associated with payments analysed in our study.

## Limitations

Our article has some important limitations. While the value of disclosed payments to practices is substantial, it excludes payments for research and development, such as clinical and non-

clinical studies, which are not disclosed on a named basis in accordance with self-regulatory rules. Moreover, we did not examine conflict of interest reporting by the practices from our dataset, which might reveal payments underreported by donors or recipients, as indicated by comparison of payments reported separately by drug companies and NHS trusts [22] and clinical commissioning groups [20] in England. Our findings are only part of a bigger picture of payments to primary care organisations. For example, companies also make payments to groups of practices or organisations involved in education of general practitioners (see examples in S3 Appendix). Extensive payments are also made to clinical commissioning groups, which procure primary care services across England [20].

Moreover, the selection of practice characteristics was not theoretically driven and omitted other potentially important ones, such as ratings of quality services. Finally, we did not examine the decision processes behind making payments nor those involved in accepting (or refusing) them. Yet, following a recent study on patient organisations, more qualitative research is needed to explore the different patterns of payments and what they might mean for practices and general practicioners and whether, and, if so, how they can influence treatment decisions [70].

## Policy recommendations

The insufficient levels of payment and conflict of interest transparency indentified by our study are concerning, particularly in relation to practices receiving substantial–in the tens of thousands of pounds–annual payments from individual drug companies. Therefore, Disclosure UK should include payment descriptions–similar to those already provided in the self-regulatory arrangements for payments to patient organisations [65]–to illuminate payments' intended goals. Similarly, without recipient identifiers data users are unable to establish the level of exposure of any practice to drug company payments [11,19]. Consequently, Disclosure UK should introduce identifiers already used by the NHS (i.e., practice codes), which would also allow for linking payment data to other publicly available datasets. Further, information about products associated with payments is necessary to investigate company marketing strategies, as is the case with the government-run US Open Payments Database [71]. In addition, comparing payments made to different HCOs requires the inclusion of recipient categories to avoid the need for checking the nature of and categorising the recipient of each payment [19]. More broadly, payments to HCOs reported in other European countries with self-regulation [6] should be itemised to allow examinining payments of different sizes.

In the long-run, a separate centralised public reporting system by practices is needed, comprising research and non-research payments from pharmaceutical and medical device companies. There are currently voluntary initiatives to make data about the links between doctors and the pharmaceutical industry publicly available, such as the UK's whopaysthisdoctor.org. However, a central register allowing patients to see the financial interest of all doctors in particular for medicines or medical devices is also being discussed [72]. The establishment of any central payment registers should be coupled with information campaigns directed at medical professionals, patients and members of the public seeking to develop their understanding of conflicts of interests. These steps seems necessary to achieve behavioural change that to-date has not been triggered by the US Open Payments database, as demonstrated by physicians' continued acceptance of COIs [73] or patients' and public's low engagement with payment data [73,74].

Beyond transparency, potential dependency of practices–or some of their activities–on drug company payments requires policy attention. The ABPI has recently acknowledged this problem by prohibiting companies following its Code of Practice from requiring being the sole

funders of HCOs [56]. Nevertheless, building on ABPI's recommendations regarding payments to patient organisations, companies should disclose the share of their payments in relevant organisational budgets [56].

## Supporting information

**S1 Appendix. Coding of general practices.**
(DOCX)

**S2 Appendix. Data cleaning protocol.**
(DOCX)

**S3 Appendix. Total value and number of payments to the top 10 healthcare organisations in England in 2015.**
(DOCX)

**S4 Appendix. Distribution of practices across different regions England.**
(DOCX)

**S5 Appendix. Breakdown of payments types received by general practices.**
(DOCX)

**S6 Appendix. Payments to the top ten general practices.**
(DOCX)

**S7 Appendix. Summary of network statistics calculated for valued networks of drug companies.**
(DOCX)

**S8 Appendix. Network visualizations for networks based on the number of payments.**
(DOCX)

**S9 Appendix. Breakdown of network statistics according to general practice characteristics.**
(DOCX)

## Acknowledgments

We would like to thank Emily Rickard for her work on categorising the recipients of drug company payments included in the Disclosure UK (2015) dataset.

## Author Contributions

**Conceptualization:** Eszter Saghy, Piotr Ozieranski.

**Formal analysis:** Eszter Saghy, Piotr Ozieranski.

**Funding acquisition:** Piotr Ozieranski.

**Investigation:** Shai Mulinari.

**Methodology:** Eszter Saghy, Piotr Ozieranski.

**Project administration:** Piotr Ozieranski.

**Resources:** Piotr Ozieranski.

**Supervision:** Piotr Ozieranski.

**Visualization:** Eszter Saghy.

**Writing – original draft:** Eszter Saghy, Shai Mulinari, Piotr Ozieranski.

**Writing – review & editing:** Eszter Saghy, Shai Mulinari, Piotr Ozieranski.

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
