## [Decision Letter · Decision Letter 0]

23 Jul 2021

PONE-D-21-20184

Drug company payments to general practices in England (2015): descriptive and social network analysis

PLOS ONE

Dear Dr. Ozieranski,

Thank you for submitting your manuscript to PLOS ONE. After careful consideration, we feel that it has merit but does not fully meet PLOS ONE’s publication criteria as it currently stands. Therefore, we invite you to submit a revised version of the manuscript that addresses the points raised during the review process.

In addition to the points raised by the reviewers, I have two additional comments:

In the Introduction I would suggest that you cite the study about the lack of disclose of industry payments to organizations sponsoring guidelines – see: Elder CMAJ 2020 June 8;192:E617-25.Are all payments to English surgeries entered into the national database that you mention on page 4?

We look forward to receiving your revised manuscript.

Kind regards,

Joel Lexchin, MD

Academic Editor

PLOS ONE

Journal Requirements:

"PO’s PhD student was supported by a grant from Sigma Pharmaceuticals, a UK

pharmacy

wholesaler and distributor (not a pharmaceutical company). The PhD work funded by

Sigma

Pharmaceuticals is unrelated to the subject of this paper.

SM’s partner is employed by PRA Health Sciences, a global Contract Research

Organization

whose costumers include many pharmaceutical companies.

ES has no conflicts of interest to declare."

Reviewers' comments:

Reviewer's Responses to Questions

**Comments to the Author**

1. Is the manuscript technically sound, and do the data support the conclusions?

Reviewer #1: Partly

Reviewer #2: Yes

2. Has the statistical analysis been performed appropriately and rigorously? 

Reviewer #1: No

Reviewer #2: I Don't Know

3. Have the authors made all data underlying the findings in their manuscript fully available?

Reviewer #1: Yes

Reviewer #2: Yes

4. Is the manuscript presented in an intelligible fashion and written in standard English?

Reviewer #1: Yes

Reviewer #2: Yes

5. Review Comments to the Author

Reviewer #1: This article addresses an interesting question and a form of institutional research funding that is unique to UK transparency reports.

General comments:

In terms of the overall framing and interpretation of study results, it would be good to know how this relates to funding of individual GPs who are working within these surgeries. Is a link possible? If not, it would be good to relate this funding to total GP funding for 2015 (if data are available) and to consider this in the overall framing / interpretation of the study results.

I am also unsure what the network analysis adds to understanding of either companies’ payment patterns or surgeries receptivity to payments. Its value to answer clear research questions on funding patterns and purpose wasn't clear. The assumption of competing companies (if they are funding the same surgery) may be incorrect, as without information about which products each company is promoting, they might or might not be competing (e.g. may be promoting in a different treatment area).

If you wanted to provide some information to readers about the pattern of company payments based on the network analysis, I would suggest just including S6(a), the figure with the network of all payments as a figure in your main report, and to explain what it shows about links between companies (ie darker lines and thicker lines mean more shared funding of the same GP surgery). Ideally you would use the concepts of 'centralisation' and 'density' as a guide for yourself for the extent of joint funding and of concentration of funding, but you would leave these jargon terms completely out of the analysis. I would leave out all of the tables and text related to this as well and just keep a very brief section on the results of the network analysis, presented in a way a general medical reader can understand. If you wanted to support this with the details of the network analysis results, this could be included in the supplement.

Detailed comments.

Page 4: please add descriptive info on surgeries - # doctors on average per surgery and extent of variation, # of GP surgeries in total in the UK, if info is available. Also the first time ‘GP surgery’ is used it should likely be translated for North American audiences, where it’s called a GP practice.

Page 5: note about the strategic importance of surgeries needs some qualification. One complication in terms of surgeries being a key target of company payments is that individual GPs working within these surgeries would be a target for payments as well. From previous research on the UK transparency data, can you say anything about the proportion of payments in total going to GPs versus to hospital-based consultants?

Page 6, methods: It’s unclear whether the focus on products that are marketed unethically will provide the information needed, as these are not necessarily the most frequently marketed products; they could be a small subset of marketed products to GPs by each company. If data are available from IQVIA or other market research companies on product-specific advertising spending in the UK over the study time period, this would be much more relevant. IQVIA has provided this type of information to academic researchers free of charge in the past (in other countries). If this information was available then a second step would be to develop a list of the subset of drugs that would be used in primary care, based on a review by a clinician or frequency of use data if available.

Page 7, methods: The decision to focus on 2015 is based on your team’s previous analysis [ref 17]. You had identified five payment categories to healthcare organisations in that study [listed in the Supplement]. Are these 5 types of payments included in the current study or a subset? Additionally eTable1 lists how the various type of health care organisations are classified in that previous article. It would be good to add a reference to how the GP surgeries were classified (for example if combined with other public sector health care providers, state this).

Page 8: You note that specific surgeries were excluded from the analysis because of an inability to match the practice code with the payment data (6.7% of total payments). I’m wondering whether all parts of your analysis required this extra information, or only specific subsets in which you were looking at the characteristics of the GP surgeries. For example, when looking at the amount each company provided to GP surgeries in total, wouldn’t it be more accurate to include the entire set?

Page 8-9: as noted above, I have some concerns about the selection of PMCPA investigations as an information source for which drugs were associated with payments to GP surgeries. You mention that this is likely the tip of the iceberg of payments. I would agree, but an additional concern is that it may not even necessarily represent the specific payment subset that focused on GP surgeries. In my opinion, this is too much of a stretch in terms of a link between this subset of payments and this subset of Disclosure UK payments. If you wanted to obtain a more accurate idea of which drugs were being most heavily promoted in 2015, my recommendation would be to try to obtain information on top drugs by company for advertising spending instead, and assess which are drugs used in primary care, as a more reliable information source on the focus of this funding of GP surgeries.

Page 9: analyses: it is interesting to examine the number of companies providing payments to each surgery and the extent to which specific companies tend to co-fund specific surgeries, which I assume is the main focus of the network analysis. I think this provides an additional very interesting focus on patterns of payments.

If I understand centralisation it measures the extent to which a set of GP surgeries with similar characteristics (regionally or by patient population?) might obtain funding from a single company. I don’t really understand this statement in terms of what it measures of importance concerning industry funding of surgeries: “Centralisation score is measured as the ratio of the actual sum of

centrality score differences and all possible sum of centrality score differences [53].”

Similarly, the measure of density as a proportion of existing ties between actors “as a share of all possible ties” needs to be explained in terms of the meaning to this specific study. It is not clear to me how this measures the “average strength of ties between companies”. It may be a situation in which companies use similar strategies to target specific GP practices, for example selecting those that have high prescribing rates and/or a patient population that is likely to use the promoted drug (e.g. if a drug for older people, a high enough proportion of older patients). Some of these marketing decisions might be directly competitive; others might simply be targeting the same surgeries for shared reasons but in a different treatment area.

P11: top of page – in describing the payments to surgeries and a proportion of total payments to healthcare organisations, it would be helpful to briefly state what types of other healthcare organisations received most of the funds. See below - the fact that they are 5th in payment level among healthcare organisations should be mentioned in results, not discussion.

Without this extra information, the latter seems inaccurate: “Consistent with our expectations, this finding suggests that surgeries were an important target of drug company payments within England’s healthcare system but their significance was reflected more by the monetary value of payments intensity of interactions, indicated by the number of payments.”

I would suggest qualifying as the number of interactions is small (0.11% of total) and monetary value needs to be seen in context, eg. likely much less than payments to individuals.

P11: fees for consultancies are presumably for the surgery to carry out consultancy services, not for individual GPs to act as consultants? Wouldn’t the latter be reported under individual payments?

Table 1: Given the large skew in the distribution of payment numbers to surgeries, I wondered whether some of this data might be better graphed than presented within a table according to quartiles. I missed being able to know actual numbers of surgeries (or proportion of surgeries with the total n provided) with different numbers of payments and/or value of payments. It would also be good to group receipt of payment by surgeries in one section (or figure) and payments by companies in another as each denominator is quite different. Denominators are also needed for numbers of surgeries and numbers of companies and it would be good to have information on the total numbers and total values of payments within the table.

I also missed information on how this subset of surgeries relates to all surgeries in the UK. What proportion have received payments? Is there a regional skew in terms of proportions of surgeries receiving payments?

P12: The following sentence is unclear: “Of the 10 companies, Bayer was clearly dominant both in relation to the number (43.41% of the top 10) and value of payments (31.96% of the top 10) and the number of surgeries to which payments were made (44.63% of the top 10).” What does ’43.41% of the top 10’ refer to exactly? Perhaps instead just say that Table 2 describes the top 10 companies by payment amount and that Bayer provided the largest amount.

Table 2 should be simplified. I would suggest leaving off most of the columns on quartiles etc. and just retaining the first 3 column on total amount paid, number of surgeries and number of payments. You could list the payment value median [IQ range] as a fourth column. Headings could be clearer and I did not understand what the percentages referred to – clearly not a percentage of total value of payments to GP surgeries. Additionally when you say ‘all HCO’ do you mean all surgeries only or all types of health care organisations? I would suggest just focusing on the surgeries in this table.

When you refer to most payments being under 869 pounds, I’d suggest instead stating the proportion under 1000 pounds or another more standard number.

Page 14 describes the proportion of surgeries receiving payments in different regions. It would have been helpful to state at the beginning of the results section what proportion of surgeries the 1790 represents. My recommendation would be to shift this to the beginning of the results section and to include the proportion per region in your table, with total numbers of surgeries per region also stated. Table 3 could easily include this information in the middle column.

An alternative would be to use the table you now call S1 in the text rather than Table 3. It provides key information – text in headings could be cut down.

Table 3: For the three sections on quartiles at the bottom, rather than saying ‘first quartile’ to 4th quartile’ you could use for example ‘Socio-economic index’ as a header row and then say ‘lowest deprivation [1st quartile] to ‘highest deprivation’ and similar for patients over 65. If these are quartiles you do not need to say numbers of surgeries per quartile. A header row ‘region’ at the top would also mean you could organize your columns differently. No need to put ‘breakdown of’ in header rows.

Page 15 – SNA is used without being defined.

Network analysis:

This entire section needs a rethink in terms of what the value to the reader of this network analysis is. I find the concepts of centralisation and density difficult to interpret in terms of the added knowledge on patterns of industry payments to GP surgeries. I can see that this is a way to measure the extent to which surgeries receive funding from many different companies and whether companies tend to have similar patterns of funding but beyond this the density and centralisation levels do not add meaning.

The description and meaning needs to be clearer. What does the high centrality score add for Bayer as compared with the data you have already presented on this company having the highest value of payments?

Additionally, the discussion of competing companies may be overinterpretation, as a company with products treating the same condition would be directly competing; others might just be targeting the same surgeries.

There appear to be different patterns by company of making many small payments versus fewer large payments. This could be a vestige of companies’ reporting patterns. Some may be more likely to break down their reports of payments to surgeries into many smaller subsets; others to amalgamate. In order to understand whether these patterns of payment size are meaningful, you would need to check the Disclosure UK reporting rules to see whether the extent of flexibility.

Our analyses of companies' payments in Australia to patient organisation suggested large differences in reporting patterns per company that were not necessarily reflective of actual differences (e.g. some seemed to amalgamate more than others). This suggested to us that value of payments per organisation was much more reliable as a measure than numbers of payments. I don't know the UK data, but if this is the case in the UK as well it should guide your analysis and interpretation.

P18: exploration of company payments and drug marketing.

I’d suggest leaving this section out as it is not directly relevant to the focus of this paper and it is a stretch to assume that these specific payments are related to the drugs with promotion that has been found to breach the industry code. A reference to the Bayer breaches for this anticoagulant (please use the generic name) might be brought in within the discussion instead, when commenting on Bayer being the company with the highest value payments to GP surgeries.

Discussion

The beginning of the discussion includes information that should be in the results. The fact that GP surgeries are the 5th type of organisation by amount of funding is important. It also refers to a table (Table S2) which states total amount of funding to GP surgeries – this funding amount also should be in the text.

Table S2 is much too long and detailed. You could stick to the top 10 organisations. Please shorten some names e.g. ‘multi-profession associations’ (4th).

P24: As noted above, the network analysis may not reflect direct drug company competition. More information is needed about which drugs are being promoted, to treat which conditions, per company, before that statement can be made. It is also possible that companies are promoting to the same surgeries for different treatment areas.

Table S3 could be simplified (e.g. only include full data rather than having two rows with full and after exclusions), but do state the denominator (# surgeries) for the table. It is useful to know the breakdown of amounts per type.

A general note: there are a number of typos in the text which need to be corrected. Secondly, most tables that include all of the quartiles plus minimum and maximum value and median could be simplified into a single column. This could be ' median [interquartile range]' or 'median [min- max]'. . An example is S4 – but this applies across the board to tables within the text and the supplement. A note for S4 – do not include the practice code unless this would be meaningful to readers.

Reviewer #2: General comments

The study/paper applies much needed scrutiny to industry payments to surgeries in England. I congratulate the authors on investigating this problem.

I have a few general comments and some specific comments below – which have all gone to both authors and editors.

The biggest suggestion I have is that the piece could be reduced in length, and at the same time made clearer. Related to this, the Methods needed to be disentangled from the Introduction and the Results need to be disentangled from the

Methods.

A statement on whether or not there was a pre-existing protocol for the study, is needed.

And an important caveat is that I have no biostats specialty, so I have not reviewed the stats.

Specific Comments

Page 4 (P4), Paragraph 1 (Pa1) You might consider referencing the recent material in BMJ, which summarises all the evidence about the distorting impacts of COIs, and flags the BMJ campaign to push for more independence. ( https://www.bmj.com/content/367/bmj.l6576 )

P5 Large paragraph. This is feeling to me like too much detail. I think you can state hypotheses much more precisely and potentially move some material to the Methods section.

P6 – First main paragraph: this feels like material that should best be in Methods.

P6 Last sentence. This sentence seems to be preempting results – as presumably you discovered this information during study – so again – feels more like methods.

P7 Methods. I would start with an overall description of the study design- before going into specifics. Also, it is important to mention whether or not there was a pre-specified protocol for the study – and if so then later describe any deviations from it.

P8, 1st paragraph. Some of this material seems to be best in results section – forgive me if I am confused, btu it feels like results are being given (in terms of numbers of pounds etc) in the middle of the Methods section. Eg this sentence “Of the total of 1,791 distinct surgeries, receiving 2,944 payments, worth £2,730,261.32, 147 (8.95%) surgeries, receiving 197 (6.69%) payments, worth £170,595.29 (6.24%), could not be linked to a practice code from the FFT dataset”

Table 2. Typo in title of Column 2: “Nnumber”

P14 typo “patters”

P14 Forgive me if I missed something – but this statement on deprived areas seems to contradict the statement below from the abstract? Unless I have misunderstood? “Contrary to our expectation that surgeries in more deprived areas will obtain more payments due to the greater number of prescriptions per patients in these regions, our data shows that surgeries in the most deprived areas (1st quartile based on MDI) received the smallest, while surgeries in the top two quartiles - the largest amount of payments”

From Abstract: “Surgeries with more patients, a greater proportion of elderly patients, and those in more deprived areas received more payments on average”

Table 3. Is there something missing from this line?

“4 th quartile 411 434.50 – 2283.12)”

P15/16/17 The material on SNA could be worded mor simply and clearly. I found the text and Table very difficult (maybe because I am Australian) – though I presume there are some important findings here – which could perhaps be explained more clearly and simply and briefly. I don’t understand at all what is being said about Tekeda on P17, and it having “more competitors”. The whole meaning and importance of “centralisation” needs to be made much clearer.

P19 – I would consider removing the detailed list of the 4 breaches, and try and summarise briefly the nature and importance of the breaches and the issues/dangers associated with the drug and/or its promotion. (I note there is relevant material in Discussion, but I would still summarise rather than list the 4)

P20 – Typo “payment patter” (unless “patter” has a meaning I am unaware of)

P24 Typo “payments”

P26. As mentioned earlier, there seems an opportunity here to be calling for more independence. It is in a way unbelievable that the NHS still allows companies to make payments to people making decisions about those companies’ drugs. The BMJ campaign could be considered, but that is up tto the authors. ( https://www.bmj.com/content/367/bmj.l6576

6. PLOS authors have the option to publish the peer review history of their article (what does this mean?). If published, this will include your full peer review and any attached files.

Reviewer #1: No

Reviewer #2: **Yes: **Ray Moynihan

---

## [Author Response · Author response to Decision Letter 0]

2 Oct 2021

We have provided responses to the Editor's and Reviewers' comments in a file forming part of this submission.

---

## [Decision Letter · Decision Letter 1]

19 Oct 2021

PONE-D-21-20184R1

Drug company payments to General Practices in England: cross-sectional and social network analysis

PLOS ONE

Dear Dr. Ozieranski,

Thank you for submitting your manuscript to PLOS ONE. After careful consideration, we feel that it has merit but does not fully meet PLOS ONE’s publication criteria as it currently stands. Therefore, we invite you to submit a revised version of the manuscript that addresses the points raised during the review process.

In addition to the comments from the reviewers, I have some additional points:

Page 3, 7^th^ line from bottom: Delete “a” after “variety”.Page 4, 7^th^ line from top: Explain the term "healthcare commissioners".Page 4, 9^th^ line from bottom: Does the term “practices” mean exclusively GP practices or does it also include specialists' practices?Page 6, last paragraph before Methods: Considering that you devote 3.5 pages in the Results to your network analysis you need to make a stronger case here about why this type of analysis is important.Page 9, first line: It should be “calculated”.Page 10, second line: I would suggest moving the sentence beginning "More frequent payments..." and the next paragraph to the end of the Introduction to help make the case why network analysis is important.Page 10, first line in Results: In the Methods you've already pointed out that your analysis is not based on 1790 practices and you've given the reason. I'd suggest eliminating this sentence and then moving up the first sentence in the next paragraph to start the Results section.Page 11, 4^th^ line: Insert “in total” after “payments”.Page 15, 6^th^ line: Is there any way of determining if companies with a shared interest are comarketing certain drugs?Page 18, 3^rd^ line from bottom: It should be “A similar trend…”Page 19, 11^th^ line from top: It's unclear what is meant by "recipients' peers".Page 19, 6^th^ line from bottom: “Medicines” is misspelled.Page 19, 2^nd^ line from bottom: It should be “covers”.Page 21, 7^th^ line from bottom: Insert “to” between “payments” and “HCOs”.Pages 22-23: I think that there is too much space devoted to the issue of unethical marketing of DOACs in general and rivaroxaban in particular, especially given the unknown relationship between rivaroxaban marketing and payments to GP practices.Page 25, 11^th^ line from bottom: Insert “a” after “Such”.Page 25, 7^th^ and 6^th^ lines from bottom: Would such a central register make a difference to patients' or doctors' behaviour? Research in the US suggests that it doesn't.Page 25, 6^th^ line from bottom: Insert “for” between “particular” and “medicines”. ==============================

We look forward to receiving your revised manuscript.

Kind regards,

Joel Lexchin, MD

Academic Editor

PLOS ONE

Journal Requirements:

Reviewers' comments:

Reviewer's Responses to Questions

**Comments to the Author**

1. If the authors have adequately addressed your comments raised in a previous round of review and you feel that this manuscript is now acceptable for publication, you may indicate that here to bypass the “Comments to the Author” section, enter your conflict of interest statement in the “Confidential to Editor” section, and submit your "Accept" recommendation.

Reviewer #1: All comments have been addressed

Reviewer #2: (No Response)

2. Is the manuscript technically sound, and do the data support the conclusions?

Reviewer #1: Yes

Reviewer #2: Yes

3. Has the statistical analysis been performed appropriately and rigorously? 

Reviewer #1: Yes

Reviewer #2: N/A

4. Have the authors made all data underlying the findings in their manuscript fully available?

Reviewer #1: Yes

Reviewer #2: Yes

5. Is the manuscript presented in an intelligible fashion and written in standard English?

Reviewer #1: Yes

Reviewer #2: Yes

6. Review Comments to the Author

Reviewer #1: Thank you for this very clear response to the previous comments and for the clarification of the social network analysis. I have identified a few very minor points of needed clarification:

Abstract:

The reference to the 'top 10 donors' does not flow that well from previous text, and the reader needs to consider this has switched from being about practices to being about companies. Perhaps instead (addition in caps):

Payments to practices were highly concentrated, AND SPECIFIC COMPANIES WERE ALSO HIGHLY DOMINANT. The top 10 donors and the top 10 recipients amassed 87.9% and 13.6% of the value of payments, respectively...

Table 1: 'value of individual payments' could be misread as being about payments to individuals. Perhaps 'value of individual payments to each practice' or 'single payment value'

Also please left justify the row labels, similarly to table 2.

page 17: this line is unclear:

"This suggests that there is an overall lower interest in the same practices as expressed by higher value payments."

Do you mean that if practices received higher value payments from one company, they were less likely to also receive higher value payments from other companies?

Please clarify.

Page 18: This line assumes too much reader recall of newly defined technical terms, especially given the similarity of terms:

"Interesting differences can be observed in terms of centralisation and centrality,"

I'd suggest :

Interesting differences can be observed in terms of centralisation (BRIEFLY REDEFINE) and centrality (BRIEFLY REDEFINE),

This introduces a little bit of repetition, but it is worthwhile for the reader.

In general, I thought that the revisions have made the analysis much clearer, and that this is an interesting article that adds substantially to the research evidence on industry payments.

Reviewer #2: In my original review I had four important main overall comments – and I do not feel that the authors have adequately addressed my comments or concerns. Obviously it is the for editors to decide whether further revision is necessary, but my sense is that it is.

My first main comment in my initial review was that the original paper was too long and could be reduced in length. The authors have responded that they have “shortened wherever possible”. I do not agree, and I think the piece is still too long. The revised Introduction alone seems to be still close to 1000 words long. The Methods section also feels too long in places, such as the data sources and social network analysis sections. It is an important study, but in my view does not warrant an exceptionally lengthy write-up.

My second main comment is that the revision should be clearer. I feel a further revision could make the whole manuscript clearer still.

My third main comment was a suggestion to try and disentangle different sections, and remove overlap between Introduction, Methods and Results. The authors responded in their letter accompanying their revision that “We have also sought to disentangle the different sections.” Yet, if I am not mistaken, in two of the three specific suggestions which I made to disentangle, they have responded that this is not possible. I think authors need to revisit my suggestions in their next revision, particularly in relation to the payment figures currently included in the Methods.

My fourth main comment, and a very important one, pertained to the issue of a pre-existing protocol. I wrote in my initial review “A statement on whether or not there was a pre-existing protocol for the study, is needed.” The authors responded to my comment by writing: “The statement about writing a protocol about the data extraction process has been added to the methods section (p. 8, para 1).” And the new revised manuscript stated “The data extraction process followed a detailed protocol which can be obtained from the authors upon request.” I am concerned because I did not suggest making a statement about a protocol for “data extraction”, I suggested making a statement about a pre-existing protocol for the study. I think any revised manuscript needs to be crystal clear about whether there was a pre-existing protocol for the study, and potentially offer an explanation for why there was not one. Moreover, if there was a pre-existing protocol for part of the study, ie the data extraction, potentially this could be made available as a supplementary file.

7. PLOS authors have the option to publish the peer review history of their article (what does this mean?). If published, this will include your full peer review and any attached files.

Reviewer #1: **Yes: **Barbara Mintzes

Reviewer #2: **Yes: **Ray Moynihan

---

## [Editor Report · Decision Letter 2]

24 Nov 2021

Drug company payments to General Practices in England: cross-sectional and social network analysis

PONE-D-21-20184R2

Dear Dr. Ozieranski,

We’re pleased to inform you that your manuscript has been judged scientifically suitable for publication and will be formally accepted for publication once it meets all outstanding technical requirements.

Kind regards,

Joel Lexchin, MD

Academic Editor

PLOS ONE

Additional Editor Comments (optional):

There are still a few minor copyediting changes that need to be made:

Page 7, 9^th^ line from bottom: Delete “using”.

Page 9, 7^th^ line from bottom: Insert “by” between “made” and “drug”.

Page 15, last line: Insert “a” between “above” and “certain”.

Page 16, first line: “Figures” is misspelled.

Page 19, 8^th^ line from bottom: Insert “we have” between “but” and “also”.
---

## [Editor Report · Acceptance letter]

29 Nov 2021

PONE-D-21-20184R2 

Drug company payments to General Practices in England: cross-sectional and social network analysis 

Dear Dr. Ozieranski:

I'm pleased to inform you that your manuscript has been deemed suitable for publication in PLOS ONE. Congratulations! Your manuscript is now with our production department. 

Kind regards, 

on behalf of

Prof. Joel Lexchin 

Academic Editor

PLOS ONE